

# A gelatin/collagen/polycaprolactone scaffold for skin regeneration

Lin-Gwei Wei[1,*], Hsin-I Chang[2,*], Yiwei Wang[3], Shan-hui Hsu[4,5], Lien-Guo Dai[6], Keng-Yen Fu[7] and Niann-Tzyy Dai[7]

[1] Division of Plastic and Reconstructive Surgery, Taoyuan Armed Forces General Hospital, Taoyuan, Taiwan, R.O.C.

[2] Department of Biochemical Science and Technology, National Chiayi University, Chiayi, Taiwan, R.O.C.

[3] Burns Research Group, ANZAC Research Institute, Concord Hospital, University of Sydney, Concord West, NSW, Australia

[4] Institute of Polymer Science and Engineering, National Taiwan University, Taipei, Taiwan, R.O.C.

[5] Research Center for Developmental Biology and Regenerative Medicine, National Taiwan University, Taipei, Taiwan, R.O.C.

[6] Department of Orthopedics, Shuang Ho Hospital, Taipei Medical University, Taipei, Taiwan, R.O.C.

[7] Division of Plastic and Reconstructive Surgery, Department of Surgery, Tri-Service General Hospital, National Defense Medical Center, Taipei, Taiwan, R.O.C.

[*] These authors contributed equally to this work.

Corresponding author
Niann-Tzyy Dai,
niantzyydai@gmail.com

## ABSTRACT

**Background**. A tissue-engineered skin substitute, based on gelatin ("G"), collagen ("C"), and poly(ε-caprolactone) (PCL; "P"), was developed.

**Method**. G/C/P biocomposites were fabricated by impregnation of lyophilized gelatin/-collagen (GC) mats with PCL solutions, followed by solvent evaporation. Two different GC:PCL ratios (1:8 and 1:20) were used.

**Results**. Differential scanning calorimetry revealed that all G/C/P biocomposites had characteristic melting point of PCL at around 60 °C. Scanning electron microscopy showed that all biocomposites had similar fibrous structures. Good cytocompatibility was present in all G/C/P biocomposites when incubated with primary human epidermal keratinocytes (PHEK), human dermal fibroblasts (PHDF) and human adipose-derived stem cells (ASCs) *in vitro*. All G/C/P biocomposites exhibited similar cell growth and mechanical characteristics in comparison with C/P biocomposites. G/C/P biocomposites with a lower collagen content showed better cell proliferation than those with a higher collagen content *in vitro*. Due to reasonable mechanical strength and biocompatibility *in vitro*, G/C/P with a lower content of collagen and a higher content of PCL ($GC_LP_H$) was selected for animal wound healing studies. According to our data, a significant promotion in wound healing and skin regeneration could be observed in $GC_LP_H$ seeded with adipose-derived stem cells by Gomori's trichrome staining.

**Conclusion**. This study may provide an effective and low-cost wound dressings to assist skin regeneration for clinical use.

## INTRODUCTION

Biocomposites, biocompatible and/or eco-friendly composites, can be formed by different varieties of natural and synthetic polymers including polysaccharides, proteins and biodegradable synthetic polymers. Generally, biocomposite materials have better structural properties than either constituent material alone. Therefore, biocomposite materials are often used in contact with living tissues such as scaffolds for cell-based therapy, biomedical implants and controlled drug delivery devices. Collagen based materials are the most well-known biocomposites in clinical use. Collagen-based wound dressings have been applied in the treatment of burn and ulcer patients over the last 30 years (*Doillon & Silver, 1986*; *Peters, 1980*). Highly innovative tissue-engineered skin substitutes have been developed to mimic normal skin recently with melanocytes, a capillary-like network, sensory innervation and adipose tissue (*Bechetoille et al., 2007*; *Regnier et al., 1997*; *Tremblay et al., 2005*; *Trottier et al., 2008*). By designing and incorporating specific therapeutic factors in skin substitutes, the promotion of wound healing as well as reduction of morbidity and mortality for large wounds may be achieved. The concept of design for the tissue engineered skin equivalent is based on existing models comprising a stratified epithelium grown on a matrix populated with dermal fibroblasts (*El-Ghalbzouri et al., 2002*). The biomaterials used for supporting skin cell growth include natural biodegradable polymers such as collagen and gelatin, as well as synthetic biodegradable polymers such as α-polyester and poly(ε-caprolactone) (PCL) (*Hajiali et al., 2011*; *Ng, Khor & Hutmacher, 2004*).

For the demand of the durability, elasticity, cosmetic appearance of normal skin and wound repair, several functional dermal layers were developed. Integra® artificial skin is a bi-layered structure consisting a dermal replacement layer covered with a silicone sheet as an outer layer. The dermal layer includes a porous fiber matrix with cross-linked bovine tendon collagen and glycosaminoglycan (chondroitin-6-sulfate) (*Burke, 1984*; *Burke, 1987*; *Burke et al., 1981*; *Yannas & Burke, 1980*). Comp Cult Skin® is a type of composite cultured skin consisting of neonatal keratinocytes and fibroblasts cultured in bovine type-I collagen scaffold. Apligraf® is a bi-layered living skin consisting an epidermal layer of neonatal keratinocytes and a dermal layer composed by neonatal fibroblasts in a type I bovine collagen gel (*Bell et al., 1981*; *Trent & Kirsner, 1998*; *Wilkins et al., 1994*).

The dermal components for most of these modified co-cultured skin constructs are mainly composed of collagen. However, pure collagen products are fragile and have difficulty in handling during clinical applications. To solve this problem, biodegradable polymers such as PCL were incorporated with collagen to increase the mechanical strength, which shows better resistance to external force for wound treatment. On the other hand, the well designed collagen scaffolds have been proven to cancel the skin wound contraction and prevent scar formation formed by contractile cells (*Dai et al., 2004*; *Dai et al., 2009*; *Powell, Supp & Boyce, 2008*; *Soller et al., 2012*; *Yannas et al., 1989*). Meanwhile, animal studies also revealed that the wound size was effectively reduced by the use of collagen *in vivo* (*Dai et al., 2004*; *Dai et al., 2009*; *Powell, Supp & Boyce, 2008*; *Soller et al., 2012*; *Yannas et al., 1989*). It may be associated with the fast growing epidermal layer of mice. In our previous study (*Dai et al., 2004*; *Dai et al., 2009*), we developed collagen:PCL (C/P) biocomposites

**Table 1  The composition and preparation of four types of GCP biocomposites.**

|  | Gelatin/ddH$_2$O (200 μl) | Collagen/1% HAc (50 μl) | PCL/DCM (500 μl) | C/G + C (%, w/w) | G + C:P (w/w) |
|---|---|---|---|---|---|
| GC$_L$P$_L$ | 2 mg/ml | 1.25 mg/ml | 7.4 mg/ml | 14 | 1:8 |
| GC$_L$P$_H$ | 2 mg/ml | 1.25 mg/ml | 18.5 mg/ml | 14 | 1:20 |
| GC$_H$P$_L$ | 0.8 mg/ml | 6.05 mg/ml | 7.4 mg/ml | 35 | 1:8 |
| GC$_H$P$_H$ | 0.8 mg/ml | 6.05 mg/ml | 18.5 mg/ml | 35 | 1:20 |
| C/P(1:8) |  | 1.85 mg/ml (250 μl) | 7.4 mg/ml |  | 1:8 |
| C/P(1:20) |  | 1.85 mg/ml (250 μl) | 18.5 mg/ml |  | 1:20 |

Notes.

G, gelatin; C$_L$, collagen with lower proportion; C$_H$, collagen with higher proportion; P$_L$, PCL with lower proportion; P$_H$, PCL with higher proportion.

in two ratios (1:8 and 1:20). They both exhibited a similar porous structure that facilitated cell proliferation. Meanwhile, the smaller pore size may prevent the direct contact between keratinocytes and fibroblasts, and allow for cell interaction via signaling through existing pores. It is worth noting that collagen-based wound dressings were expensive. To solve this problem, we chose gelatin, the degradation of collagen, for the preparation of skin biocomposites. Therefore, the purpose of this study was to prepare a cost-effective, mechanically strong and biodegradable biocomposites based on gelatin, collagen, and PCL (G/C/P). Moreover, G/C/P biocomposites with a lower collagen content were designed and tested in this study because of good cell attachment and proliferation in our preliminary testing. Therefore, high and low GC mats with two different ratios of PCL were fabricated. The structure, thermal characteristics and biocompatibility were evaluated by in vitro testing. The potential in promoting large-sized wound healing was confirmed in the full-thickness skin defect model of nude mice.

## MATERIALS AND METHOD

### Preparation of G/C/P and C/P biocomposites

The preparation of the G/C/P and C/P biocomposites was listed in Table 1. Aliquots of type B gelatin (from bovine skin, Sigma-Aldrich, St Louis, MO, USA) with specific concentration of collagen (Sigma-Aldrich, St Louis, MO, USA) solution was prepared by dissolving gelatin in double distilled water followed by mixing collagen (dissolved in acetic acid) to produce two ratios of collagen in gelatin/collagen biocomposites respectively. The dissolution of gelatin/collagen was facilitated by stirring using a heating magnetic stirrer in a 25 ml glass shell vial at the temperature of 40 °C. After complete dissolution, aliquots (0.25 ml) of the gelatin/collagen solution were added to 7 ml glass vials followed by elimination of air bubbles and frozen at −20 °C for approximately 45–50 min. In the second stage, samples were transferred to a freezer at −72 °C for 35 min. The frozen samples were placed in a freeze dryer (Edwards Modulyo®) at −44 °C under 42 mbar vacuum for 24 h to prepare for the G/C mats. Aliquots (0.5 ml) of PCL (Mw=115000; Solvay Interox, Warrington, UK) /dichloromethane (DCM with HPLC grade; Fisher Scientific, Loughborough, UK) solution were added carefully to the freeze dried G/C mats with low collagen content to produce GC$_L$P$_L$(G/C:PCL is 1:8) and GC$_L$P$_H$ (G/C:PCL is 1:20) biocomposites respectively, whereas

aliquots (0.5ml) of PCL/dichloromethane (DCM) solution were added to the freeze dried gelatin/ collagen mats with high collagen content to produce $GC_HP_L$ (G/C:PCL is 1:8) and $GC_HP_H$ (G/C:PCL is 1:20) biocomposites respectively. The vials were kept for 30 min and then removed the lids to allow solvent evaporation overnight.

On the other hand, C/P biocomposite was fabricated described by previous study (*Hajiali et al., 2011*). Briefly, collagen solution was prepared in 1% acetic acid by stirring with a magnetic stirrer overnight at room temperature. Aliquots (0.25 ml) of the collagen solution were added to 7 ml glass shell vials (Fisher Scientific) and frozen at –20 °C for approximately 45–50 min. Samples were transferred to a freezer at –70 °C for 35 min before freeze drying. Aliquots (0.5 ml) of a solution of PCL in DCM were added carefully to the freeze dried collagen mats to prepare 1:8, 1:20 collagen:PCL materials, respectively. The vials were kept stopped for 30 min before removing the lids to allow solvent evaporation overnight. The component ratios of the four G/C/P and C/P biocomposites are shown in Table 1.

## Protein release from G/C/P biocomposites

To determine the release of gelatin and collagen from biocomposites ($n = 3$), the bicinchoninic acid (BCA) assay was used to estimate the amount of protein release from G/C/P biocomposites after incubation in phosphate-buffered saline (PBS) at 37 °C for 12 days. Individual samples were added to 7 ml glass shell vials containing 1 ml PBS and then incubated at 37 °C in a water bath. The release media was replaced completely by fresh PBS periodically and analysed for total protein content using the BCA assay. The absorbance of the calibration samples measured at 562 nm was used to produce a calibration curve used to calculate the protein (gelatin/collagen) concentration of the test samples.

## Thermal analysis by differential scanning calorimetry (DSC)

The thermal characteristics of $GC_LP_L$, $GC_LP_H$, $GC_HP_L$, and $GC_HP_H$, biocomposites with weight between 2–10 mg were recorded using a Perkin-Elmer Pyris Diamond differential scanning calorimeter. All the samples were lightly pressed into the bottom of the pan to ensure good thermal contact. Sealed DSC pans were used in the study. Triplicate samples were heated at a rate of 10 °C/min from 10 °C to 100 °C. Peak melting temperature (Tm) and heat of fusion data for the PCL component of the materials were determined using the built-in software of the DSC. The latter measurement was subsequently used to estimate the percentage crystallinity of PCL in the composites from the reported heat of fusion of 139.5 J/g for fully crystalline PCL (*Burke, 1987*). Indium was used as a standard.

## The tensile strength analysis by a universal testing machine

The mechanical analysis was conducted on GCP and C/P biocomposites using a universal testing machine (Instron) under axial loading. The biocomposite specimens were cut as the suitable size of rectangular block and were carefully clamped at the center of the cross-head with its end faces exactly perpendicular to the longitudinal axis. The crosshead speed of 50 mm/min was applied for this test. The tensile strength (MPa) was calculated as the force at failure divided by the cross-sectional area of the biocomposite. Results were the

average from 3–6 measurements and analyzed by one-way analysis of variance and the $t$-test ($P < 0.05$).

## Observation of porous structure by scanning electron microscopy (SEM)

The scanning electron microscope was used to observe the surface morphology of G/C/P biocomposites. Samples were attached to aluminum SEM stubs using carbon tabs (Agar Scientific). Specimens were sputter coated with gold prior to examination using a HITACHI®S-3000N scanning electron microscope.

## Cell culture on G/C/P

Primary human epidermal keratinocytes (PHEK) and primary human dermal fibroblasts (PHDF) were isolated and primarily cultured from the donated human foreskin samples after the surgery of circumcision in the study, which were approved by the institutional review board (IRB). The study protocol was reviewed and approved by the Institutional Review Board (IRB) in the Tri-Service General Hospital, R.O.C. (TSGHIRB No.: 100-05-251). Then the written informed consent was obtained from each donor. EpiLife® HKM (Cat No.M-EPIcf-500 with addition of 0.06 mM calcium chloride and HKGS kit: Cat No.S-001-5) was used for keratinocyte culture. The cell isolation from skin samples, cell expansion, and cell counting followed those described previously (*Burke, 1984*).

PHEK (child foreskin; P4) were seeded on the top surface of G/C/P (1:8 and 1:20) and C/P biocomposites and 24-well tissue culture plastics (TCP) at a cell density of $1.7 \times 10^5$ or cells per cm$^2$ for up to 9 days, PHDF (adult foreskin; P3-4) were seeded on the same materials at a cell density of $2.0 \times 10^4$ cells per cm$^2$ for up to 10 days. Human adipose-derived stem cells (ASCs; passages 3–4) were seeded on the biocomposites and 24-well TCP at a cell density of $2.0 \times 10^4$ cells per cm$^2$ for up to 12 days. Trypsin-EDTA solution was used for cell detachment followed by cell counting at 1, 3, 6, and 9 days for PHEK and cell at 1, 4, 7, and 10 days for PHDF using a Weber's haemocytometer. The experiments were repeated at least twice with similar results.

The adhesion, growth, and distribution of cells seeded on G/C/P biocomposites were investigated using the immunohistochemical assay. PHEK (child foreskin; P4) were seeded on the top surface of G/C/P biocomposites (1:8 and 1:20), C/P biocomposites and 24-well tissue culture plastics (TCP) at a cell density of $1.7 \times 10^5$ cells per cm$^2$ for time intervals up to 9 days. PHEK on G/C/P biocomposites (1:8 and 1:20) at 1 and 3 days were labeled with the (primary) monoclonal rabbit anti-human cytokeratin (CK)-14 1:100 (Bioworld Technology), and the (secondary) rhodamine-conjugated polyclonal goat anti-rabbit IgG 1:200 (Chemicon, Billerica, Mass) examined under a fluorescent microscope.

PHDF (adult foreskin; P3-4) were seeded on the top surface of 1:8 and 1:20 GC10, GC25 and collagen:PCL biocomposites and TCP (24-well tissue culture plastics) at a cell density of $2.0 \times 10^4$ cells per cm$^2$ for time intervals up to 10 days. PHDF on G/C/P biocomposites (1:8 and 1:20) at 1 and 3 days were labeled with the (primary) monoclonal mouse anti-human α-tubulin antibody 1:200 (Cat No.sc-5286; Santa Cruz Biotechnology, Inc.), and the (secondary) fluorescein (FITC)-conjugated polyclonal goat anti-mouse IgG

1:100 (Lot No.62686; Jackson ImmunoResearch Laboratories, Inc., Baltimore, MD, USA) and examined under the fluorescent microscope.

## Preparation of human adipose-derived stem cells (ASCs)-seeded G/C/P biocomposites for animal experiments

Human adipose tissues were obtained using lipoaspirate after syringe-assisted liposuction from abdomens of adult Taiwanese female patients who had no systemic metabolic diseases or lipid disorders. The procedures and protocols were conducted at the Tri-Service General Hospital, Taipei, Taiwan, and were approved by IRB.

The isolated fat tissues were washed using PBS. After washing, the tissues were digested with an equal volume of PBS including 0.075% type I collagenase (Sigma-Aldrich Company Ltd, Poole, UK) at 37 °C for 1 h. After centrifugation at 2,500 rpm for 10 min, the cell pellet was resuspended in Dulbecco modified Eagle medium-low glucose (DMEM-LG; Invitrogen) containing 10% fetal bovine serum (FBS; Invitrogen) and 1% penicillin-streptomycin (Invitrogen), and placed in the incubator. After 1 day of culture, the dishes were washed with Hanks buffered salt solution (HBSS; Thermo Scientific, Agawam, MA, USA) to discard blood cells and nonadhesive cells and fresh culture medium was then added. The medium was refreshed every 3 days. When the cells reached 70% to 90% confluence, the cells were trypsinized (0.25% trypsin; Sigma), neutralized with cultured medium and then passaged at a ratio of 1:3. Cells of passages 3 to 4 were used in the experiments.

G/C/P biocomposites were cut to fit 12-well tissue culture plastics and then placed into the wells of TCP. For each well, human adipose-derived stem cells (ASCs) at a cell density of $10^6$ cells per cm$^2$ were seeded onto the biocomposite with the addition of cultured medium. After incubation for 24 h, the ASC-seeded biocomposite was washed with HBSS to remove the culture medium and nonattached cells.

## Animal model and wound healing experiments

A skin wound healing model of nude mice was employed (*Huang et al., 2012*). All mice were acquired, housed, and studied under a protocol approved by the Institutional Animal Care and Use Committee of National Defense Medical Center, Taiwan, R.O.C. The three eight-week-old nude mice (BALB/c-nu; BioLASCO) were chosen for each group including $GC_LP_H$ scaffold only and $GC_LP_H$ scaffold seeded with ASCs in this study. All of the surgical instruments were sterilized and the surgical procedures were performed under laminar flow. The surgical sites were sterilized with Easy Antiseptic Liquid 2% (Panion & BF, Taipei, Taiwan) before surgery. After anesthesia, a square of full-thickness cutaneous wound (12 mm ×12 mm) was made by surgery using scalpel on each dorsum of the hind thighs, followed by grabbing, pulling the region with a forceps, and excising of the full-thickness tissue with scissors.

The wounds were divided into the following two groups: blank $GC_LP_H$ biocomposite and ASCs-seeded $GC_LP_H$ biocomposites. The $GC_LP_H$ biocomposites (0.1–0.2 mm thick) for ASCs seeding were used. Both grafts were placed on each wound, sewn with 6 to 8 stitches using NC125L Nylon 5-0 surgical sutures (UNIK, Taipei, Taiwan), and covered with Tegaderm films (3M Health Care, MN) to prevent catching, biting, or wound infections.
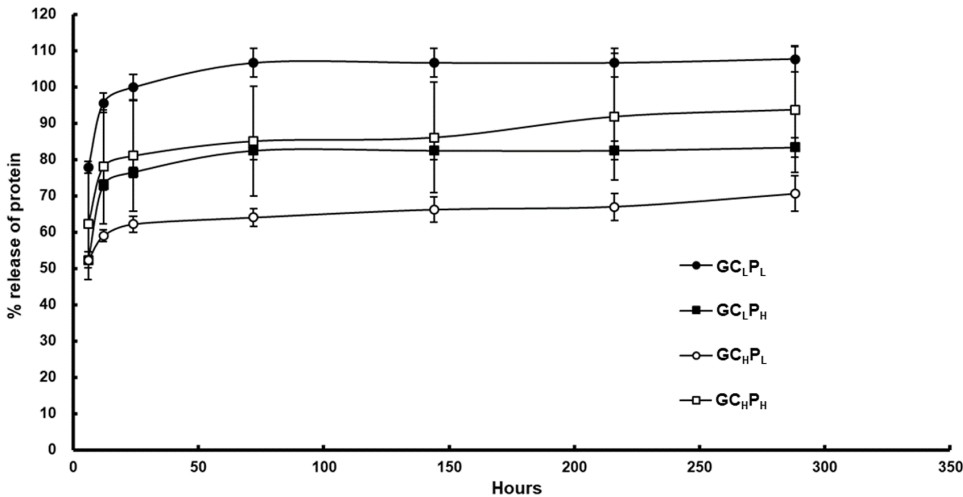

**Figure 1** Cumulative protein release from G/C/P biocomposites in PBS at 37 °C.

The wounds were continuously observed for a period of 3 weeks. The area of the wounds was blindly measured twice using ImageJ v1.44p software (http://imagej.nih.gov/ij; NIH, Bethesda, MD, USA). After 2 weeks, the tissues of healed wounds were excised, fixed in 10% formalin for at least 24 h at room temperature, subsequently embedded in paraffin, and sectioned in 5-μm increments. The sections were stained with Gomori's trichrome staining and examined by an optical microscope (Olympus BX50, Hamburg, Germany) (*Luna, 1992*). The collagen component of the extracellular matrix deposited in skin substitutes was stained green.

## Statistical analysis

Results were presented as the mean ± standard deviation of three replicates for each experiment. Statistical analysis was performed using Statistical Package for the Social Sciences, Version 12.0 (SPSS Inc., Chicago, Illinois). The statistically significant differences between groups were assessed by one-way ANOVA analysis of variance, followed by Tukey post-hoc test. $P < 0.05$ was considered statistically significant.

## RESULTS

### Protein release from the G/C/P

The compositions and fabrication of G/C/P biocomposites are present in Table 1 and the cumulative release of protein from the biocomposites is shown in Fig. 1 and Table 2. For G/C/P with a lower collagen content, a higher cumulative protein release was present for $GC_LP_L$ biocomposite in comparison with $GC_LP_H$ group. At 6 h, the former showed about 78% average release, whereas the latter showed 52% average release. The protein was released quickly in 12 h for both biocomposites but slow down afterwards. The cumulative release (w/w) at 24 h was 100% and 76% for two biocomposites respectively. For G/C/P with a higher collagen content, a higher cumulative release of protein was found for $GC_HP_H$ biocomposite relative to $GC_HP_L$ biocomposite. The former exhibited a higher

**Table 2  Cumulative protein release from G/C/P biocomposites in PBS at 37 °C.**

|  | Gelatin (%) | Collagen (%) | PCL (%) | 6 h (%) | 12 h (%) | 24 h (%) | 72 h (%) | 144 h (%) | 216 h (%) | 288 h (%) |
|---|---|---|---|---|---|---|---|---|---|---|
| $GC_LP_L$ | 9.61 | 1.50 | 88.89 | $78 \pm 2$ | $96 \pm 3$ | $100 \pm 4$ | $107 \pm 4$ | $107 \pm 4$ | $107 \pm 4$ | $108 \pm 4$ |
| $GC_LP_H$ | 4.12 | 0.64 | 95.24 | $52 \pm 2$ | $73 \pm 1$ | $76 \pm 1$ | $82 \pm 3$ | $82 \pm 3$ | $82 \pm 3$ | $83 \pm 3$ |
| $GC_HP_L$ | 7.99 | 3.12 | 88.89 | $52 \pm 1$ | $59 \pm 2$ | $62 \pm 2$ | $64 \pm 2$ | $66 \pm 3$ | $67 \pm 4$ | $71 \pm 5$ |
| $GC_HP_H$ | 3.42 | 1.34 | 95.24 | $62 \pm 15$ | $78 \pm 16$ | $81 \pm 15$ | $85 \pm 15$ | $86 \pm 15$ | $92 \pm 18$ | $94 \pm 17$ |

Notes.

G, gelatin; $C_L$, collagen with lower proportion; $C_H$, collagen with higher proportion; $P_L$, PCL with lower proportion; $P_H$, PCL with higher proportion; h, hours (time).

**Table 3  The analysis of G/C/P biocomposites for thermal characteristics ($n = 3$, values are expressed as mean $\pm$ SD).**

|  | Tm (°C) | Crystallinity (%) | Reference |
|---|---|---|---|
| $GC_LP_L$ | $59.2 \pm 1.5$ | $53.9 \pm 1.8$ |  |
| $GC_LP_H$ | $61.5 \pm 2.0$ | $54.8 \pm 2.4$ |  |
| $GC_HP_L$ | $58.3 \pm 0.2$ | $55.1 \pm 1.2$ |  |
| $GC_HP_H$ | $59.8 \pm 0.3$ | $57.1 \pm 1.4$ |  |
| PCL (100%) film | $62.8 \pm 0.2$ | $54.2 \pm 5.4$ | *Dai et al. (2004)* |
| C/P (1:8) | $60.7 \pm 0.2$ | $48.5 \pm 4.4$ | *Dai et al. (2004)* |
| C/P (1:20) | $63.7 \pm 0.4$ | $63.5 \pm 3.8$ | *Dai et al. (2004)* |

average release rate of about 62% average release at 6 h, compared to 52% average release of the latter. The average cumulative release (w/w) after 24 h was 81% and 62% respectively for $GC_HP_H$ and $GC_HP_L$ biocomposites. The cumulative release reached a plateau for both biocomposites after 24 h. PCL is the polymer with slow degradation rate and low water uptake and hence $GC_LP_H$ showed better collagen and gelatin encapsulation due to high PCL content. Therefore, *unencapsulated* gelatin or collagen were released within 3 days and the remained gelatin or collagen would be released slowly depending on the PCL degradation rate. In addition, the amount of protein release significantly increased by gelatin amount in G/C/P biocomposites with low PCL content of polymer but no effect in the those with high PCL content of polymer.

## Thermal and tensile analyses for G/C/P biocomposites

The thermal properties of G/C/P biocomposites ($n = 3$) are listed in Table 3. For G/C/P made with $GC_L$ mats, the melting point of the PCL component was close to the value of 60 °C normally found for pure PCL. The crystallinity of the PCL in these biocomposites (C/P and G/C/P) was estimated by comparing the fusion heat with that of 100% crystalline PCL as—139 J/g. From our previous DSC study (*Dai et al., 2004*), the 1:20 collagen:PCL biocomposite displayed the highest crystallinity of $63.5 \pm 3.8\%$, followed by $48.5 \pm 4.4\%$ for 1:8 collagen:PCL biocomposites. However, the crystallinity of the PCL films prepared by solvent casting was only around $54.2 \pm 5.4\%$ in comparison with 1:20 w/w collagen:PCL biocomposites at $63.5 \pm 3.8\%$ (Table 3). For G/C/P biocomposites, the average crystallinity of $GC_HP_H$ was found to increase by 2.9% relative to the solvent cast PCL film. Furthermore,

**Table 4** The tensile properties of four GCP and C/P biocomposites ($n = 3$; values are expressed as mean ± SD).

|  | Young's modulus (MPa) | Elongation (%) | Tensile strength (MPa) |
|---|---|---|---|
| $GC_LP_H$ | 35.5 ± 8.3 | 10.7 + 1.8 | 2.5 ± 1 |
| $GC_HP_H$ | 46.6 ± 6.2 | 6.7 ± 1.7 | 2.2 ± 0.4 |
| $GC_LP_L$ | 30.8 ± 9.1 | 5.1 ± 1.5 | 0.8 ± 0.1 |
| $GC_HP_L$ | 59.5 ± 8.3 | 7.4 ± 1.8 | 2.2 ± 0.5 |
| C/P (1:20) | 75.8 ± 4.2 | 9.4 + 1.3 | 3.7 + 0.4 |
| C/P (1:8) | 5.7 + 3.8 | 18.1 + 6.1 | 0.5 ± 0.1 |

the mean crystallinity of the PCL phase in the higher gelatin/collagen content (1:8 w/w) tended to be lower than that of 1:20 films. However, there was no statistically significance between these groups (Paired $t$-test: $P = 0.173$).

The mechanical properties of G/C/P vs. C/P are listed in Table 4. The enhanced tensile strength and elongation and lower Young's modulus were detected in $GC_LP_H$ when compared with C/P biocomposites (1:20 mixture).

## Structure of G/C/P biocomposites

The SEM images of G/C/P biocomposites are shown in Fig. 2. From these image, an irregular pore structure (20–100 µm) was apparent on the surface of G/C/P biocomposite. The biocomposites with a lower PCL content ($GC_LP_L$) exhibited more porous structure. On the other hand, biocomposites with a higher collagen content ($GC_HP_L$ and $GC_HP_H$) showed less pore structure when compared to those with a lower collagen content ($GC_LP_L$ and $GC_LP_H$). The $GC_HP_L$ biocomposite with highest content of collagen and lowest content of PCL showed a rough, fibrous surface due to the underlying collagen mat structure. The higher ratio of PCL was applied, the smoother surface was present because smooth overlying areas could be formed by the PCL phase. In addition, the surfaces of biocomposites with higher ratio of collagen seemed to have gel-like material filling the pores. All biocomposites exhibited the rough, fibrous structure of G/C mat overlaid by the smooth PCL phase. Little difference in morphology was observed.

## Attachment and proliferation of PHEK, PHDF and ASC on G/C/P biocomposites

The attachment and proliferation of PHEK, PHDF and ASCs on G/C/P are shown in Fig. 3. The number of PHEK on $GC_LP_H$ and $GC_HP_L$ was greater than that on the other biocomposites at 1 day. The cell density of PHEK was greater on C/P biocomposites (1:8) than on G/C/P biocomposites at 3 and 6 days ($P < 0.05$). PHEK almost had equal cell density on C/P and G/C/P biocomposites at 9 days ($P > 0.05$). Moreover, the $GC_LP_L$ biocomposite appeared to have the greatest number of cells at 9 days. However, no significant differences were present among these groups. The number of PHDF on $GC_HP_L$ was greater than on the other biocomposites after 1 day ($P < 0.05$). There was no significant difference in the cell number among all groups at 4, 7, and 10 days. Results of cell proliferation on different biocomposites showed that PHEK exhibited faster cell

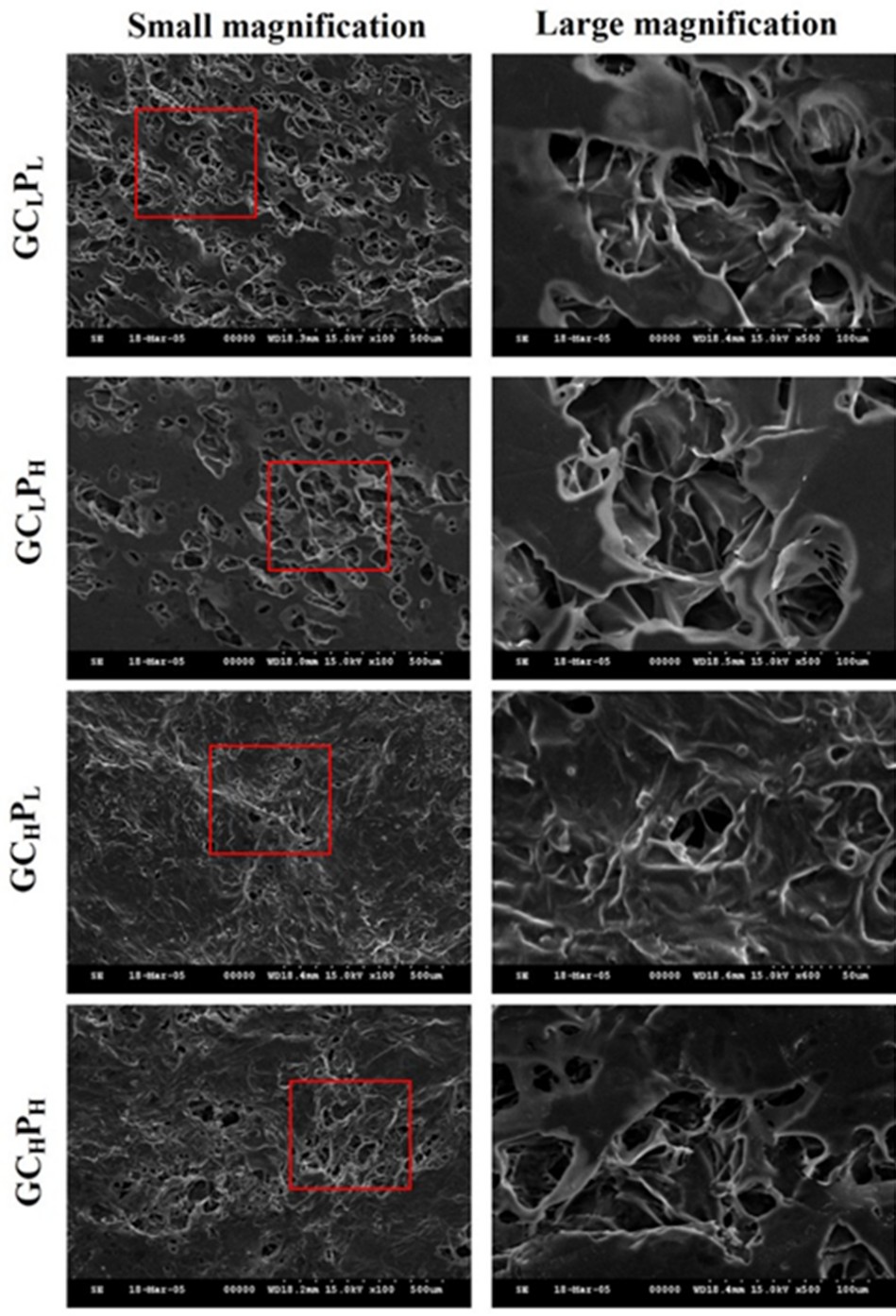

**Figure 2** SEM images of $GC_LP_L$, $GC_LP_H$, $GC_HP_L$, and $GC_HP_H$ biocomposites.

growth than PHDF at 1 day. Meanwhile, the numbers of ASCs on different biomaterials were also investigated. Cell proliferation on $GC_LP_H$ was similar to the other groups in 10 days. Based on these results, PHEK, PHDF and ASCs well attached and proliferated on all G/C/P biocomposites.

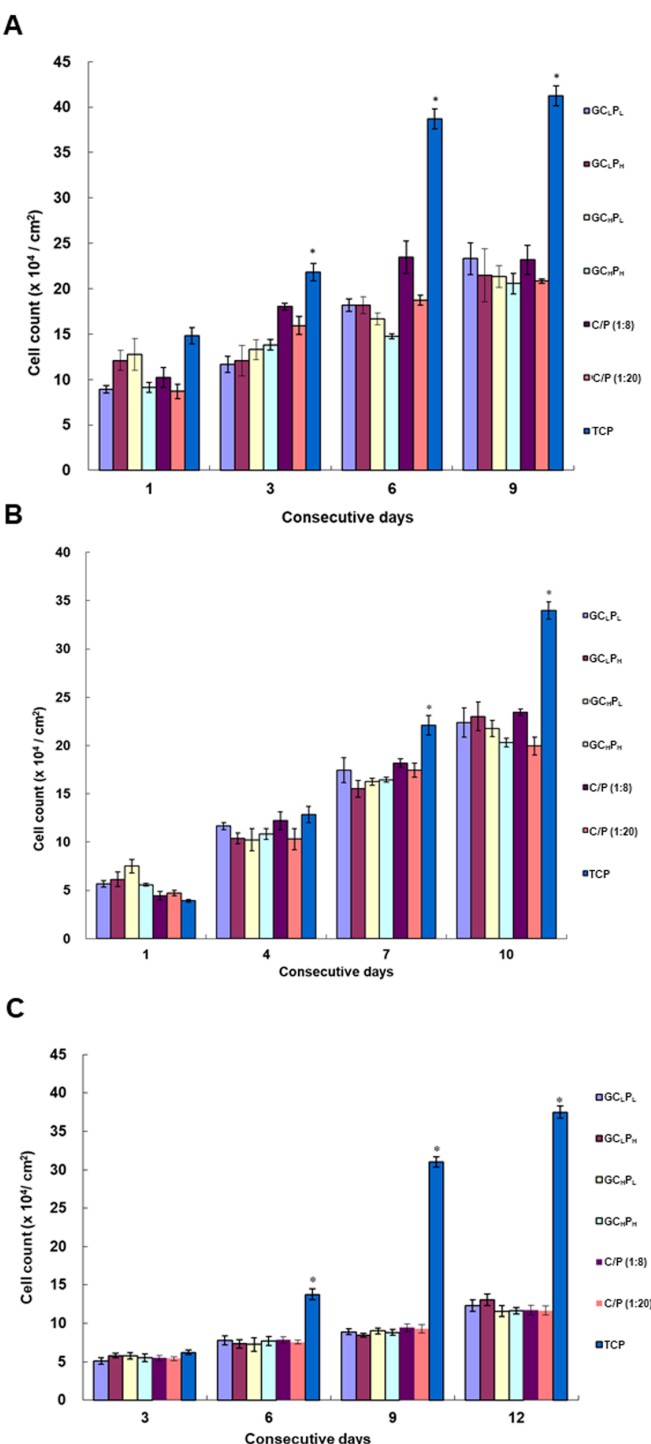

**Figure 3** **Attachment and proliferation of (A) Primary human epidermal keratinocyte (PHEK) and (B) primary human dermal fibroblasts (PHDF) (C) human adipose-derived stem cells (ASCs) grown on G/C/P and C/P biocomposites.** The blank wells (tissue culture polystyrene, TCP) served as the control. ∗: indicates $P < 0.05$ between TCP and other groups.

The morphology of PHEK and PHDF is shown in Fig. 4. Keratinocytes with similar granular shape were observed on the biocomposites. A higher expression of cytokeratin-14, the structural protein of mature keratinocyte, was induced as the incubation time increased. On the other hand, α tubulin, a major component of fibroblast cytoskeleton, was stained and appeared as the spindle-like shape. As the time increased, more α tubulin was expressed in PHDFs. The fluorescent expressions of cytokeratin-14 and α tubulin were parallel to the numbers of attached and proliferated cells shown in Fig. 3. In addition, the α tubulin expression of fibroblasts and cytokeratin-14 expression of keratinocytes seeded on $GC_LP_H$ and $GC_HP_L$ were higher than those of other groups at three days. Therefore, the $GC_LP_H$ biocomposite with the lowest collagen content in the 4 GCP scaffolds was selected as the skin substitute for the pilot animal study because gelatin was a cheaper material than collagen.

### In vivo animal study

Rapid wound closure was observed in the both groups covered with $GC_LP_H$ biocomposites in comparison with the control group, as shown in Fig. 5. In both groups covered with $GC_LP_H$ biocomposites, the wound size dramatically decreased apparently was observed from 8 to 12 days and then gradually decreased until 21 days. The rate of wound healing was almost the same for both groups within 16 days. However, the wound healing based on wound area exhibited statistical difference at day 21 ($P < 0.05$). The incomplete wound closure also observed even up to 21 days for the group covered with $GC_LP_H$ only but not ASC-seeded $GC_LP_H$. Furthermore, Gomori's trichrome staining was performed in control, groups covered with $GC_LP_H$ only and ASC-seeded $GC_LP_H$, as presented in Fig. 6. The open wound covered with $GC_LP_H$ only exhibited loose collagen deposition even after 14 days (Figs. 6B & 6E). In the ASC-seeded $GC_LP_H$ group, complete wound closure with differentiated epidermis and abundant dermal parallel-arranged fibrous collagen deposition was observed at 14 days (Figs. 6C & 6F). At a larger magnification, the ASC-seeded $GC_LP_H$ group showed the largest thickness of epidermis, followed by the $GC_LP_H$ only group, and then the open wound. The thick epidermis layer was obviously observed in the outside portion of the wound for the ASC-seeded $GC_LP_H$.

## DISCUSSION

Based on the lower collagen content, we developed a cheaper skin substitute integrating the advantages of natural and synthetic biopolymers to promote the growth of skin cells in this study. Berillis has mentioned that collagen-based biopolymers are critical for tissue engineering and regenerative medicine due to its superior biocompatibility and low immunogenicity which is depended on the source of collagen (*Berillis, 2015*). Collagen not only provides the building block for elastin and collagen fiber formation, but also acts as ligands for dermal fibroblasts to stimulate the production of new collagen, elastin and hyaluronic acid (*Sibilla et al., 2015*). In addition, collagen hydrolysate exhibits bioactivities including antihypertensive activity, lipid-lowering activity, as well as reparative properties in injured skin (*Fan, Zhuang & Li, 2013*). Gelatin, which is derived from the partial hydrolysis of collagen, could be effective in promotion of granulation and epithelialization in the skin

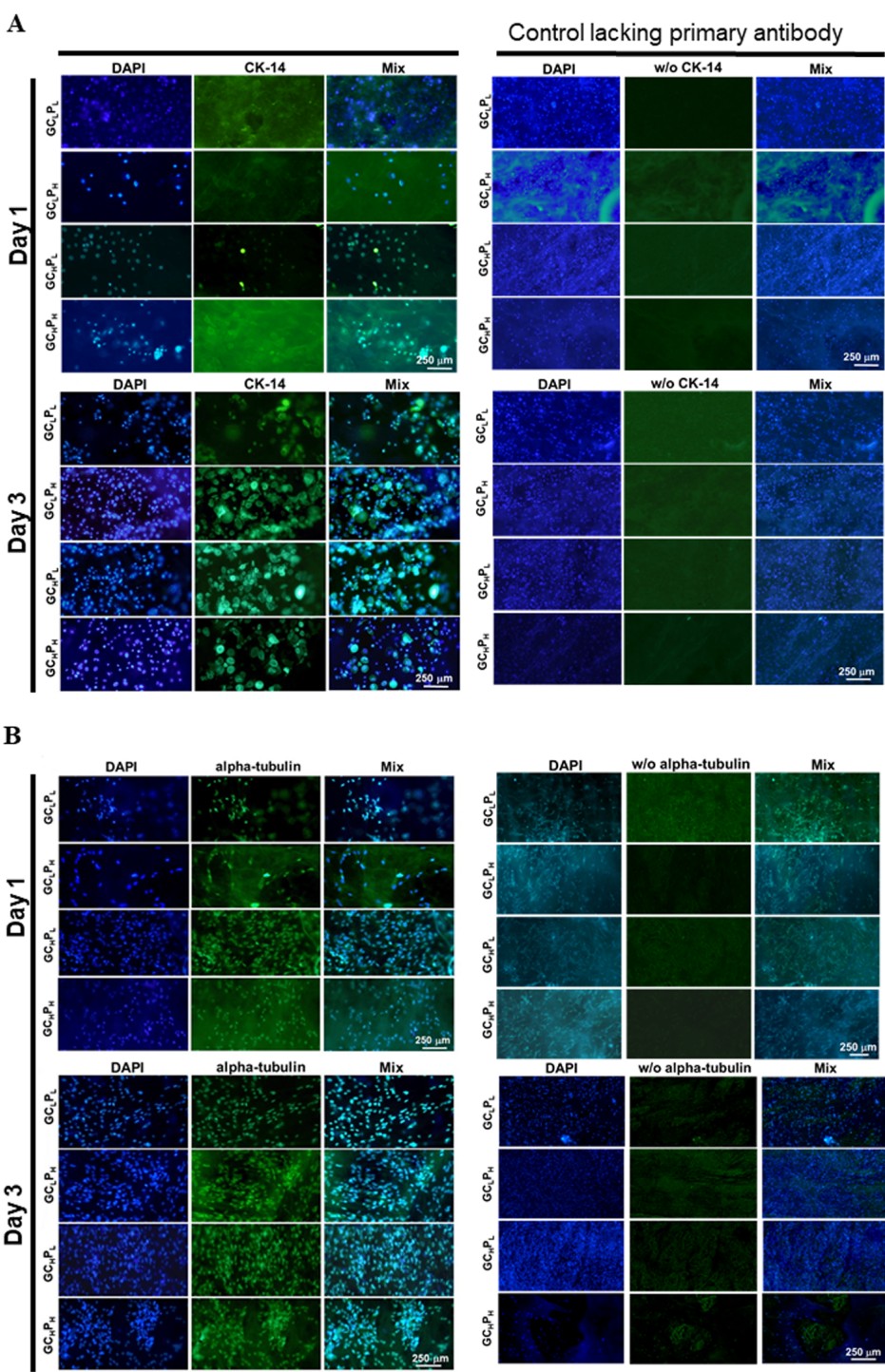

**Figure 4** **Immunohistochemical assay for (A) PHEK grown on four G/C/P biocomposites at 1, 3 days and for (B) PHDF grown on four biocomposites at 1, 3 days, as the staining controls lacking primary antibody were present in the second set of columns.** CK-14 was a specific protein for PHEK, whereas alpha tubulin was a specific protein for PHDF. w/o, without. (scale bar: 250 μm).

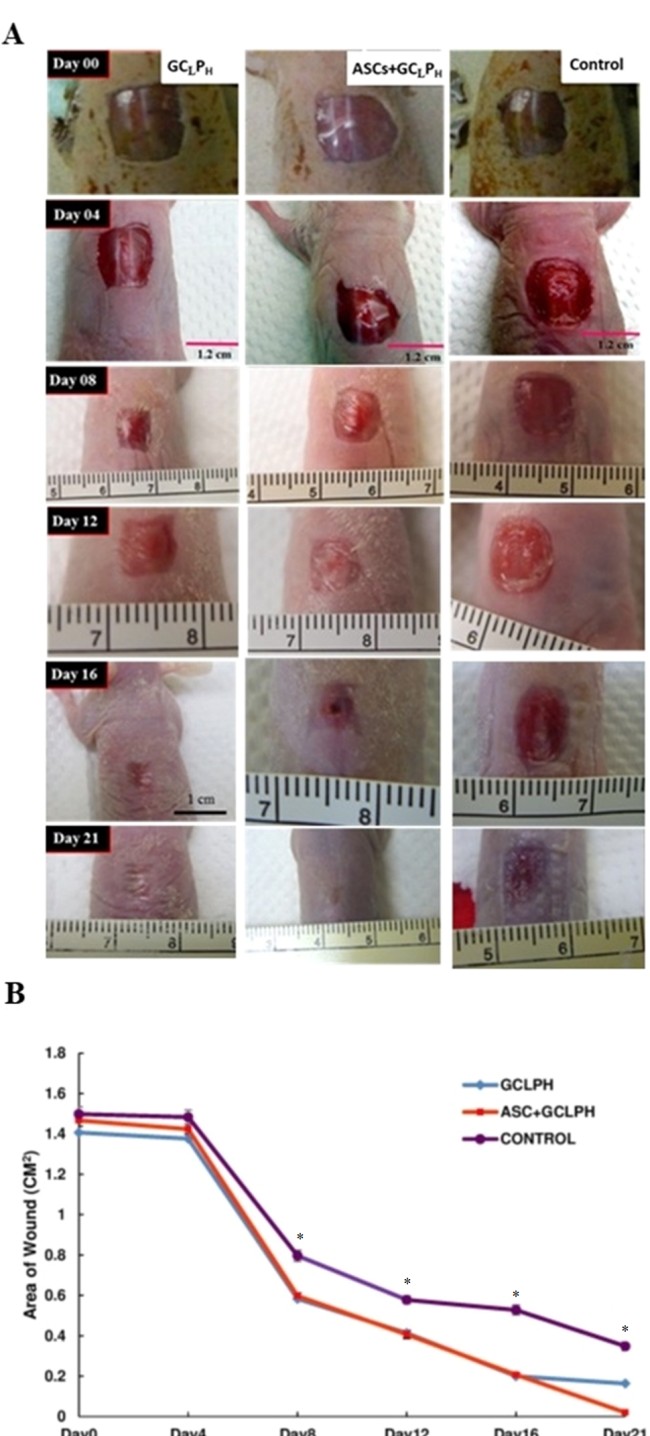

**Figure 5** **Comparison of wound closure: (A) the appearance and (B) the area of wound in the full-thickness skin defect of nude mice covered with GC$_L$P$_H$ scaffold and those seeded with ASCs until 21 days.** The untreated wounds were served as the control. ∗: indicates $P < 0.05$ between control and other groups.

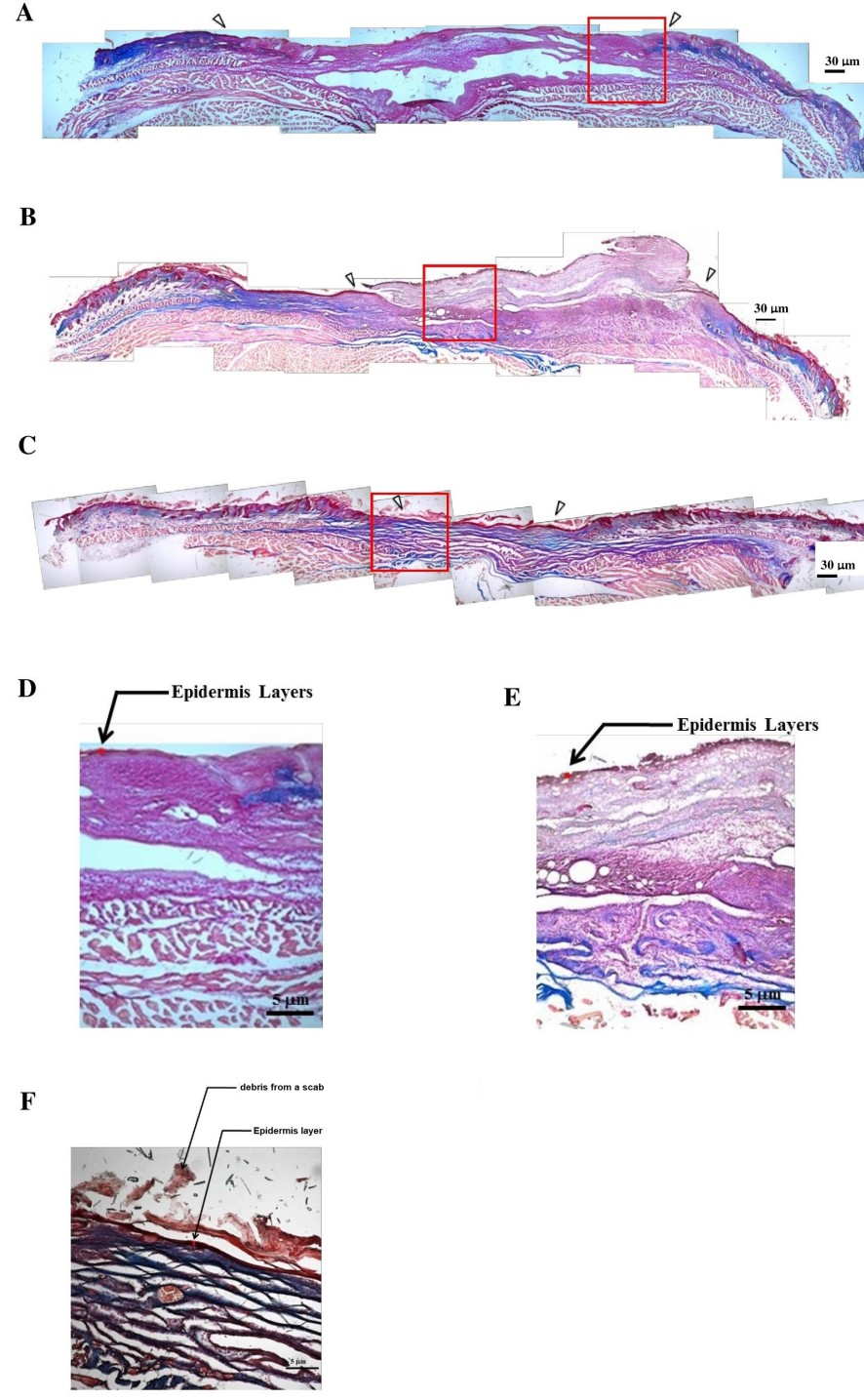

**Figure 6 Collagen deposition in the wound bed by Gomori's trichrome staining for the histology of cultured skin model.** Collagen deposition in the wound bed by Gomori's trichrome staining for the histology of cultured skin model in the groups of (A) open wound (B) $GC_LP_H$ only (C) ASCs-seeded $GC_LP_H$ (40×; scale bar: 30 μm) on the full-thickness skin defect of nude mice for 14 days. (D) to (F) were the magnified images of the box area (red rectangle) in (A) to (C) respectively (100×; scale bar: 5 μm).

wound and also modulate drug delivery in targeted tissues (*Albuquerque-Jr et al., 2009*; *Dantas et al., 2011*; *Ikada & Tabata, 1998*). Gelatin contains integrin binding sites for cell adhesion with the characteristics of low cost, natural abundance, biodegradability and biocompatibility (*Lakshminarayanan et al., 2014*; *Ma et al., 2005*). Poly- ε-caprolactone (PCL), approved by the Food and Drug Administration (FDA), is a hydrophobic, biodegradable and biocompatible synthetic polyester and which are widely applied in drug delivery and tissue repair (*Spearman, Rivero & Abidi, 2014*). Therefore, our approach could combine the characteristics including cell binding properties of natural polymers such as gelatin and collagen and the structural stability of PCL. These composite films are highly flexible which may allow effective draping over the wound site to improve the graft application.

Tissue engineering scaffolds should have enough residence time for tissue development. PCL was chosen in this study because of its long resorption time, over 1 year (*Pitt et al., 1981*). Additionally, the stability of PCL reduces the possible toxicological effect associated with chemical crosslinking of natural polymers. Tear resistance is also conferred by PCL which facilitates the manipulation during surgery. We also expect the stability of PCL may inhibit wound contraction by fibroblasts for optimal dermal regeneration of full thickness wounds (*Werner et al., 1994*). However, PCL is known to be susceptible to acid and enzymatic degradation (*Leffler & Muller, 2000*). *Mochizuki et al. (1995)* reported that certain lipases enhanced the degradation of polycaprolactone (PCL) when compared with incubation in buffer only.

Based on DSC data, the degree of PCL crystallinity was greater for the biocomposites with a lower G/C content in which implicated that the gelatin/collagen phase could impede PCL crystallization. The notable reduction of PCL crystallinity in the biocomposites with $GC_H$ mat may be associated with the effect of $GC_H$ mat to interfere with the nucleation of PCL crystals. This phenomenon may also facilitate hydrolytic chain scission in this semi-crystalline polymer. In addition, the solvent evaporation may be conducive to develop high crystallinity due to the differences of crystal nucleation process between the pellet and the thin membranous form of PCL. Gelatin may not be able to develop a very strong architecture during the process of lyophilisation thereby exerts a minor effect on crystal nucleation of PCL phase. From DSC results, the similar GC content had almost the same percentage of crystallinity in PCL (Table 1). Therefore, collagen content may play a critical role in the degree of the PCL crystal nucleation pattern.

As the PCL content decreased, C/P biocomposites showed a more open, porous structure in a previous study (*Coombes et al., 2002*; *Dai et al., 2004*). In previous study, the cell attachment and spreading on collagen:PCL composites found that 1:8 composites exhibited a greater masking or coating effect of the fibrous structure, whereas 1:20 composites showed a roughened, dimpled surface texture and separated pores. A similar trend in G/C/P was observed based on SEM images of this study. The pore number and size were slightly greater for the lower collagen content. Our previous study reported that the pore number and size seemed to increase after incubation in PBS, suggesting that protein may be released from the sub-superficial layers of the biocomposites (*Dai et al., 2004*). As shown in Fig. 1, $GC_HP_L$ and $GC_LP_H$ remained higher gelatin/collagen amount than other two G/C/P biocomposites

after 24 h incubation and Fig. 3 also indicated that $GC_HP_L$ and $GC_LP_H$ had slightly better cell attachment than others. Therefore, cell attachment and proliferation may be more related the amount of remained gelatin or collagen rather surface morphology.

The cost effect, quality of materials, and simple fabrication procedures are critical for the development of tissue-engineered skin model. Although collagen is more bioactive than gelatin, the price for collagen is much higher than that of gelatin. Therefore, the replacement of collagen by gelatin or other biomaterials may be cost-effective. However, pure gelatin was relatively fragile and weak during the lyophilization procedure, which made the fabrication difficult to control and caused one-third failure preparation rate in the early stage of this work. From Table 4, the C/P biocomposite also showed lower tensile strength, lower elongation, and higher Young's modulus when compared with $GC_LP_H$ biocomposites. Therefore, the mixture of gelatin and collagen as G/C mat was employed. The G/C/P biocomposites were found to possess proper biocompatibility.

ASCs are widely used in tissue engineering and stem cell therapy because of their advantages in accessibility and immunosuppressive characteristics (*Gonzalez-Rey et al., 2009*). Effective would healing was improved by ASCs in literature (*Lam et al., 2013*; *Lin et al., 2013*; *Nauta et al., 2013*; *Shokrgozar et al., 2012*). The adipose lineage cells could serve as a new cell source that promoted reepithelialization and angiogenesis in dermal wound healing from previous literature (*Ebrahimian et al., 2009*). However, stem cells directly applied on the wound site are not well survived (*Lam et al., 2013*), which reduces the efficacy of wound closure. An abundant amount of dermal collagen deposition was shown by histological staining. These findings suggested that G/C/P biocomposites are good carrier for ASCs. To verify the fate of adipose stem cells seeded, a tracing dye has to label the cells, which will be a future subject of study. In addition, G/C/P biocomposites may further be incorporated with growth factors to accelerate the wound repair (*Shin & Peterson, 2013*).

Finally, we hypothesize that the stronger mechanical strength of G/C/P may help resisting the contraction generated by fibroblasts and prevent possible scar formation. Therefore, the G/C/P biocomposites (in particular $GC_LP_H$) with advantages in biocompatibility as well as mechanical properties are potential scaffolds for clinical wound treatment.

## CONCLUSIONS

G/C/P biocomposites were fabricated and characterized as tissue engineered skin substitutes in this study. A similar fibrous/porous structure and thermal properties were present in G/C/P biocomposites. A reasonable mechanical strength and biocompatibility *in vitro* were achieved for the G/C/P biocomposites containing the lower collagen content ($GC_LP$). Furthermore, the rapid closure of the skin wounds in nude mice were observed for those treated with $GC_LP_H$ only or ASC-seeded $GC_LP_H$. Therefore, $GC_LP_H$ biocomposites could be applied as low cost wound dressings based on high content of inexpensive gelatin and reasonable healing capacity.

### Funding

The financial support for this work was provided by the Ministry of National Defense, ROC (MAB-106-034), National Defense Medical Center, Tri-Service General Hospital, ROC (TSGH-C106-112), Zouying Branch of Kaohsiung Armed Forces General Hospital, ROC (ZBH 105-07), Taoyuan Armed Forces General Hospital, ROC (AFTYGH-104-27, AFTYGH-105-29), Ministry of Science and Technology, ROC (MOST-102-2314-B-016-009, MOST-103-2314-B-016-013) and Ministry of Economic Affairs (ROC) program, grant number 98-EC-17-A-19-S2-0090. Additional financial support was provided by Teh-Tzer Study Group for Human Medical Research Foundation, ROC. The funders had no role in study design, data collection and analysis, decision to publish, or preparation of the manuscript.

### Grant Disclosures

The following grant information was disclosed by the authors:
Ministry of National Defense, ROC: MAB-106-034.
National Defense Medical Center, Tri-Service General Hospital, ROC: TSGH-C106-112.
Zouying Branch of Kaohsiung Armed Forces General Hospital, ROC: ZBH 105-07.
Taoyuan Armed Forces General Hospital, ROC: AFTYGH-104-27, AFTYGH-105-29.
Ministry of Science and Technology, ROC: MOST-102-2314-B-016-009, MOST-103-2314-B-016-013.
Ministry of Economic Affairs (ROC) program: 98-EC-17-A-19-S2-0090.

### Competing Interests

The authors declare there are no competing interests.

### Author Contributions

- Lin-Gwei Wei, Hsin-I Chang and Yiwei Wang analyzed the data, authored or reviewed drafts of the paper, approved the final draft.
- Shan-hui Hsu contributed reagents/materials/analysis tools, prepared figures and/or tables, approved the final draft.
- Lien-Guo Dai and Keng-Yen Fu performed the experiments, approved the final draft.
- Niann-Tzyy Dai conceived and designed the experiments, approved the final draft.

### Human Ethics

The following information was supplied relating to ethical approvals (i.e., approving body and any reference numbers):

The study protocol was reviewed and approved by the Institutional Review Board (IRB) in the Tri-Service General Hospital, R.O.C. (TSGHIRB No.: 100-05-251).

### Data Availability

Figshare: https://figshare.com/articles/A_gelatin_collagen_polycaprolactone_scaffold_for_skin_regeneration/5938159.

## Supplemental Information

Supplemental information for this article can be found online at http://dx.doi.org/10.7717/peerj.6358#supplemental-information.

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
