# Peer review of "A gelatin/collagen/polycaprolactone scaffold for skin regeneration"

_PeerJ, doi:10.7717/peerj.6358_

## Round 0.1 · original submission · Major Revisions

Please address the comment from two reviewers. In your revision, please also clarify the advances/contributions of this study.

·

Basic reporting

no comment

Experimental design

no comment

Validity of the findings

no comment

Additional comments

A Review of:

Title: A gelatin/collagen/polycaprolactone scaffold for skin regeneration
This study investigated the effect of composition and formulation of gelatin/collagen/PCL polymeric materials on in vivo efficacy to enhance skin wound healing, especially with the aid of stem cell incorporation. Although a series of material characterizations was evaluated for various compositions, material’s functionality to facilitate in vitro cellular proliferation and in vivo skin tissue regeneration does not seem to be significant. Specifically, in order to conclude that this biocomposite material sufficiently support full thickness in vivo skin regeneration, authors should have included the experimental control group such as sham. All other comments listed below.
1. Please revise any typos, grammatical errors, and abbreviations.
2. Provide how statistical analysis was performed in the method section.
3. Figure 1 (Protein release): It might be better to provide a degradation profile of biocomposites in the in vitro condition with the release of gelatin and collagen from the materials. Total release amount of gelatin/collagen, evaluated by conventional BCA assay, should be correlated with that of mass loss and physiological degeradation of biocomposites over time.
4. Figure 2 (Surface morphology of fabricated biocomposites): It would be more informative to provide any discussion about a surface coverage (as seen in Fig 2) of the fabricated scaffold materials and any possible inhibition of cellular infiltration. In general, the topography/ surface morphology and correated overall porosity might influence cell and tissue infiltration in an in vivo environment.
5. Figure 3 (Cellular proliferation on the top of resulting scaffolds): Please remark any statistical difference between formulations in the figure. Moreover, please elaborate discussion about how G/C/P formulation affect the in vitro cellular proliferation.
6. Figure 5 (in vivo functionality): Again, it is hard to conclude that G/C/P scaffold could facilitate skin would healing, unless the sham control is provided. Please also remark statistical difference in Figure 2 (Was the only significance observed on Day 21?).

·

Basic reporting

Clear english- yes
Literature - insufficient
Structure - acceptable, I find it better to discuss results as they are presented and not in a separate section but authors followed PeeJ standard
Figures - Not all are professional (Fig 4)
Raw data - Not all is supplied
Self contained- yes, but with limitations

Experimental design

Yes
Yes-
Yes-
Yes-

Validity of the findings

Data is not robust
Conclusions are not well supported

Additional comments

Development of more effective wound healing procedures and substitutes is a clinical need, but it is my opinion that with this work the authors did not achieve a significant advancement and conclusions are not well supported by the data.

L29- Bioplastics – definitely not a commonly used adjective to characterize polymers like gelatin, collagen and polycaprolactone. Not even the authors use it on a regular basis, the final appearance of bioplastics is on line 72, when the paper has 424. Moreover, the work contextualization, first 10 lines of the introduction, in totally off, the world use of petroleum-based plastics has nothing to do with a biomedical application of gelatin, collagen and polycaprolactone.

L111- “both biocomposites” – did the authors mean “both types of biocomposite”? , because 6 different composites were produced that can be grouped in two sets

L111- Please confirm that type B gelatin is of bovine skin origin

L155- Please check that the cross-head speed was 50 mm/min. This is a very high speed for a mechanical test, why did the authors choose it and the consequences must be discussed.

L157 – 3 measurements is insufficient given the typical variability of test results given the fluctuations in sample production and preparation.

Table 1- The way authors choose to report solution composition is in no way reader-friendly. For instance, 0.0004g/0.2ml should be reported as 2 mg/ml. This is totally different in terms of readability. Also, final percentage of each polymer in the biocomposites should be given as this simplifies the interpretation of results like protein release.

Fig1- Protein release is very fast, therefore the usefulness of having a gelatin/collagen component in the scaffold should be discussed. Also, why, given the “burst release” of the protein component the authors did not opt to stabilize this component by crosslinking it.

Fig 1, Line 243 and others- Since protein release exceeds 100% in the case of GCLPL, this means that the calibration curve may be significantly offset. Given this uncertainty, it is incorrect to state release values up to the tenth percent.

Fig 1- the authors do not give any explanation on why for some scaffolds protein loss is total whereas for others it is not. The authors simply describe their factual results without attempting an explanation. Is protein loss correlated with gelatin content? Is collagen more resistant to dissolution? Is collagen lost by day 3, or does it also dissolve? Why should the GCLPL scaffold lose all its G/C content and the GCHPL retain about 30%?

L247 – Why should scaffolds that are very similar behave so different in terms of protein loss? Why should they plateau by day 3 at different release values? Does plateau in reality mean total release?

L256– The original DSC/TGA spectra should be supplied as supplementary information.

L259- I am not sure about the explanation given for the different crystallinity values. Crystallinity seems to increase with PCL content, which leads me to ask the authors if the melting enthalpy, which the software calculates given the sample mass, was corrected for the G/C component.

L263- The 1% difference is irrelevant, smaller than the experimental uncertainty, which is more than just a standard deviation.

L265- J/g

L278- Why didn’t the authors determine mechanical properties for all scaffolds?

L278- “tensile strength” should be compressive strength, “elongation” should be strain and elastic or compressive modulus would be better than Young’s modulus.

Table 3- Uncertainties should be expressed with one digit only (not including zeroes to the left).

L284 and Table 2- Pore size cannot be determined from SEM images, period. The definition of pore in such structures is by no means straightforward. Measurement of a “pore” area is highly prone to bias. If techniques like mercury intrusion or bubble point porosimetry had been used, operator bias would have been removed and everyone knows about limitations and approximations with these methods. Here, we know nothing about the measurements because authors did not supply annotated SEM images.

L294 – Authors do not report how many times these experiments were performed and with how many replicas each time. Given the overall behavior of cell populations with time, I do not see any significantly different cell behavior on all samples tested. Differences reported are within the experimental uncertainty given the variability of these experiments.

Fig 4 has no quality. We only see blurred images impossible to interpret. Moreover, controls without primary antibody and DAPI counterstaining should have been performed.

L318 – Given that only the scaffold GCLPH was tested for its mechanical properties, how can it be selected in detriment of others based on its “proper mechanical properties” (why are they proper?)

L223- In the experimental methods the authors state that: “The wounds were divided into the following two groups: blank GCLPH biocomposite and ASCs-seeded GCLPH biocomposites”. Fig 5 reports results for these two groups, but then Fig 6 contains data for two additional groups.

L 326- “wound contraction was evidently observed only in the GCLPH group at 16 and 21 days” – Wound contraction happens in both groups and appears to be higher for the ASCs + GCLPH group.

Fig 6 – Is “Area of wound “ the area of the wound that has not been re-epithelialized? If it is, how did the authors evaluate this area?

L327 – “Gomori’s trichrome staining was performed in the ASC-seeded GCLPH, open wound, ASCs-only, and non-seeded GCLPH groups” – The open wound and ASCs-only groups were not reported until this sentence!

L338 – The novelty of this skin substitute is very limited. The relevant literature, where very similar scaffolds were studied, is not comprehensively discussed.

L348 – It is not true that gelatin has no drawbacks. Fast dissolution is one.

L365 – PCL is degraded by enzymes like collagenase, MMPs and so on?

L420 – There was no optimization, simply the choice of a scaffold with the lowest collagen content.

L423- “rapid closure and re-epithelialization of skin wounds were observed in nude mice treated with ASC-seeded GCLPH biocomposites” - an important phenomenon with clinical implications is wound contraction and scarring, which were not discussed.

L424- No advances were demonstrated in comparison with currently used treatment methods.

---

## Round 0.2 · Major Revisions

There are still some serious issues from the previous round of review that have not been addressed in your current response, e.g. statistical analysis from Reviewer #1 and several material-related questions from Reviewer #2. Please address the concern and questions from two reviewers in your response. This is a final opportunity to address the concerns of Reviewer 2 in a suitable way

·

Basic reporting

Statistical analysis is missing in so figures and relevant discussions are necessary.

Experimental design

According to #1 (Basic reporting), it is necessary to report any significance in formulations to conclude the therapeutic efficacy of the present materials.

Validity of the findings

N/A

Additional comments

N/A

·

Basic reporting

Clear english- ok
Literature - acceptable
Structure - ok
Figures - Fig 4 is still difficult to observe
Raw data - ok
Self contained- not really: authors want to lower collagen content in scaffolds they studied previously but replaced collagen by gelatin that was not cross linked so it dissolved pretty fast and authors ended up studying several PCL scaffolds.

Experimental design

Yes
Yes- : research question is relevant but methods used do not lead to a convincing result.
Yes- : I have no reason to have doubts about technical and ethical standards, it's experimental planning that, to me, was not the best
Yes

Validity of the findings

Data is not robust, statistically weak
Conclusions are still not well supported

Additional comments

In spite of the corrections and improvements made by the authors, this new version still requires attention to quite a few issues detailed below:

L30 – The last mention of “bioplastics” still standing

L31 – The methods description became more confusing than before. My suggestion: “G/C/P biocomposites were fabricated by impregnation of lyophilized gelatin/collagen (GC) mats with PCL solutions, followed by solvent evaporation.
Two different GC:PCL ratios (1:8 and 1:20) were used.”

L40 – remove “Besides,”

L42 – “than those”

L45 – “As expected” may be an indication of some bias result analysis, not the best way to start a description of the main results achieved.

L48 – The scaffolds the authors developed are not just a “Wound dressing”, they are aimed at assisting/promoting skin regeneration.

L256 - Protein release from the G/C/P: the results are still presented without convincing explanations (those put forward by authors do not explain all results) and in a difficult to read manner. Quantitative results should be presented in a table with composite composition (percentage of each polymer in the scaffold) alongside.

- Also, the authors didn’t answer my question: “Why should the GCLPL scaffold lose all its G/C content and the GCHPL retain about 30%?”.

L258 - The authors argue that differences in protein loss between GCLPL (higher loss) and GCLPH (smaller loss) are due to the higher PCL content of GCLPH that encapsulates G and C. But then, why is it that GCHPL that has a smaller PCL content also has a smaller protein loss that the GCHPH?

L262 – “The cumulative release (w/w) at 24 hours was 100% and 82.6% for two biocomposites respectively. The released rate reached a plateau for both biocomposites after 3 days. “ – If the cumulative release reaches 100% after 24h, then the plateau is reached after 24 h, not after 3 days… Also, instead of “release rate”, “cumulative release” should be used.

L284 – The authors didn’t directly answer my question, if the melting enthalpy, which the software calculates given the sample mass, was corrected for the G/C component. However, the values they now present are higher than those in the preceding version, which suggests this correction was made. Is this the reason why values are different?

L284 – A crystallinity of 84.7% for PCL is something I don’t recall ever seeing reported. Also, I don’t understand to which scaffold this value pertains as the text reads “the 1:20 collagen:PCL biocomposite displayed the highest crystallinity of 84.7%, followed by 58.7% for 1:8 collagen:PCL biocomposites”, are these Dai et al (2004) scaffolds or are these the authors present scaffolds? Moreover, these numbers are not on table 2 that still contains the same numbers as the previous version.

L285 – In the previous version, the crystallinity of PCL films prepared by solvent casting was characterised as “notably low at 54.2 ± 5.4%” but now it is “unrealistically high at 54.2±5.4%”… what happened?

L291 – In my previous report I questioned the authors “Why didn’t the authors determine mechanical properties for all scaffolds?”. The answer, “We measured the G/C/P composites” does not really answer my question because the authors produced 4 G/C/P composites and only one, which was later used in the in vivo tests, was tested here, denoting a bias towards this scaffold.

L307 – Given the production method used and the fact that most, if not all, protein content was lost (I wonder what the advantage of incorporating one was), cells are attaching to a PCL surface. Therefore, no significant differences should show up in day 1 populations and the differences reported may be simply due to the difficulty in detaching and collecting all cells for subsequent counting using trypsin. What surprises me most is that fibroblasts, which are easy cells to culture and attach, seem to have had a great difficulty in attaching to TCP, an unusual behavior.

Fig 4 – The images I have access to are the same as in the previous submission, blurred and with some of the arrows pointing to almost dark areas. Also, without staining controls lacking primary antibody and without DAPI counterstaining there is no way to know if the arrows are pointing at cells, dirt or some other feature. I wonder how authors could measure the areas reported, what their criteria was.

L604 – “Immunohistochemical”

L328 – “involucrin expression of keratinocyte on GCLPH was only slighter lower than that of GCHPL – From Fig 4C it is about two thirds, which, in my opinion, does not qualify as “only slighter lower”. In fact, from fig 4A, the images pertaining to GCLPH exhibit the faintest staining.

L334 – Wounds heal by a combination of contraction, which is clearly seen in Fig 5, synthesis of a new dermis and a new epidermis, also called re-epithelialization. Wound contraction is easy to observe but re-epithelialization is not and at least from the figures the reader cannot distinguish between open wound (without epidermis) and closed wound (with epidermis). The authors did not answer my question: how did they evaluate re-epithelialization?

L334 – Why is it that in Fig 5A one cannot visually confirm the presence of the scaffold? Wounds receiving the scaffold and control look alike. The scaffold should impart a whitish appearance to the wound.

L343 – “We could observe that the biocomposite (GCLPH only) became incorporated into the regenerated skin” – How? What do figures show that can be unambiguously identified as the biocomposite? What is identified in Fig 6F as biocomposite is simply a scab.

---

## Round 0.3 · Minor Revisions

Please address the remaining minor concern from the two reviewers.

·

Basic reporting

It is hard to understand which comparisons were performed in Figure 3 (in vitro proliferation of cells). Even though authors stated "∗: indicates P<0.05 relative to control" (i.e., TCP) in the Figure caption 3, the statistical difference is indicated in the control group itself. It seems that any significance is observed in the control group as compared with other formulations. Please revise them all.

In addition, more importantly, the statistical analysis should be also performed in the Figure 5 (in vivo wound closure). In order to emphasize the therapeutic efficacy of the present platform, any significant difference in wound closing area should be indicated as compared with the untreated wound control (purple line), at each given time point.

Experimental design

no comment

Validity of the findings

no comment

Additional comments

no comment

·

Basic reporting

All OK

Experimental design

All OK

Validity of the findings

All OK

Additional comments

The manuscript was improved and may be published after a few minor issues are solved:

- L187 “immunohistochemical”

- L280 – I still don’t understand why the crystallinity of PCL films is “unreallistically high at 54%” when the crystallinity of 1:20 collagen:PCL composites is higher, at 63%

- L321 – alpha-tubulin dimerizes with beta-tubulin to form microtubules, a major component of fibroblast cytoskeleton, but not “the main component”

- L326 – The GCLPH composite was chosen for its collagen content, the lowest of the 4 PGC scaffolds studied, not because it “showed the nearest tensile strength to 1:20 C/P biocomposites”, this is irrelevant.

- L603 and 665 – What are “positive cells”?

- L631 – The SD should be presented with only one, or at the most two, digits (not counting zeros to the left of the first non-zero digit)

- L653 – The meaning of the second set of columns (4, 5 and 6) of figure 4 is not stated, nor what is Anti-GFP, nor why there are repeated images in Fig 4.B

-L661 – Fig 6F has an arrow pointing to what is supposedly a piece of the biocomposite but that looks like debris from a scab.

---

## Round 0.4 · accepted · Accept

The authors made the changes requested by the reviewers.

#